# Dual-function enzyme acts as a global c-di-GMP sink and local anti sigma factor antagonist to drive cellular differentiation

Buse Cınar Cakmak[1], Johanna D. Saric[1], Katrin Wrede[1], Neil A. Holmes[2], Julian Haist[3¤], Maria A. Schumacher[4], Mark J. Buttner[2], Natalia Tschowri[1]*

1 Institute of Microbiology, Leibniz Universität Hannover, Hannover, Germany, 2 Department of Molecular Microbiology, John Innes Centre, Norwich, United Kingdom of Great Britain and Northern Ireland, 3 Department of Biology/Microbiology, Humboldt-Universität zu Berlin, Berlin, Germany, 4 Department of Biochemistry, Duke University School of Medicine, Durham, North Carolina, United States of America

¤ Current address: Tilray Deutschland GmbH, Densborn, Germany
* tschowri@ifmb.uni-hannover.de

## Abstract

Soil bacteria of the genus *Streptomyces* are natural producers of over two-thirds of clinically used antibiotics. Their ability to generate these valuable metabolites is tightly linked to a developmental program involving the transition from vegetative hyphae to spores. The second messenger cyclic di-GMP (c-di-GMP) stabilizes effector complexes that block sporulation, including the RsiG-σ$^{WhiG}$ complex leading to sequestration of the developmental sigma factor by its anti sigma factor. How signal termination and disruption of effector complexes is achieved to allow sporulation, remains poorly understood. Here, we identify the phosphodiesterase RmdB as a dual-function regulator that terminates c-di-GMP signaling both globally and locally. We show that deletion of the *rmdB* gene leads to increase of the global c-di-GMP pool and delayed development. Using genetic complementation, we demonstrate that both the EAL motif and the GGDEF domain are essential for the physiological function of RmdB. Our co-immunoprecipitation and co-elution assays revealed that RmdB interacts directly with the sigma factor σ$^{WhiG}$ via its GGDEF domain, thus preventing binding of the anti sigma factor RsiG to σ$^{WhiG}$ and promoting sporulation. Our bacterial two-hybrid analyses identify RmdB as an interaction hub connecting to multiple diguanylate cyclases (DGCs), including CdgE, which also interacts with σ$^{WhiG}$. These findings establish a novel principle of bacterial signaling in which a phosphodiesterase serves as an antagonist of an anti sigma factor, integrating global second messenger degradation with local effector complex formation to control cell fate decisions.

**Data availability statement:** All relevant data are within the manuscript and its Supporting Information files.

**Funding:** This work was supported by the European Research Council under the European Union's Horizon 2020 research programme (grant 101039556, SecMessFunctions, to N.T.); by zukunft.niedersachsen, the joint science funding program of the Lower Saxony Ministry of Science and Culture and the Volkswagen Foundation to N.T.; and by the National Institutes of Health (R35GM130290 to M.A.S.). The funders had no role in study design, data collection and analysis, decision to publish, or preparation of the manuscript.

**Competing interests:** The authors have declared that no competing interests exist.

## Author summary

Bacteria use small signaling molecules to coordinate complex behaviors. In the filamentous soil bacterium *Streptomyces venezuelae*, the signaling molecule cyclic di-GMP controls the switch from growing hyphae to spore formation, a developmental transition that is closely linked to antibiotic production. High cyclic di-GMP levels prevent sporulation by stabilizing protein complexes that keep developmental regulators inactive. However, how this signal is removed at the right place and time has remained unclear. In this study, we identify the enzyme RmdB as a key factor that shuts down cyclic di-GMP signaling in two ways. First, RmdB reduces the overall cellular level of cyclic di-GMP by degrading the molecule. Second, RmdB acts locally by binding the developmental sigma factor WhiG through its GGDEF domain and interfering with the anti sigma factor RsiG, which normally keeps WhiG inactive. This reveals that RmdB is not only a signal-degrading enzyme, but also a direct regulator of a developmental protein complex. Our findings uncover a new principle of bacterial signaling in which one protein integrates global signal removal with local control of effector activity to drive cell differentiation.

## Introduction

Nucleotides fulfil many different functions essential for cellular life. They are crucial building blocks of nucleic acids, represent important players in energy transfer and energy storage reactions, act as co-enzymes and are often employed as second messengers in signalling networks of cells. Nucleotide-based second messengers are small, diffusible molecules which can be composed of single nucleotides, such as 3′,5′-cyclic adenosine monophosphate (cAMP) and 3′,5′-cyclic guanosine monophosphate (cGMP) or two nucleotides, for example bis-(3′-5′)-cyclic dimeric guanosine monophosphate (c-di-GMP) and bis-(3′-5′)-cyclic dimeric adenosine monophosphate (c-di-AMP) [1]. Classical signalling cascades leading to an adaptive reaction of a cell as a response to a stimulus often involve a well-characterized order of events: (I) perception of a signal leading to activation of a second messenger synthase and production of the molecule, (II) binding of the signalling molecule to an effector (III) modification of the effector by changing its activity or conformation leading to a physiological response, and (IV) degradation of the second messenger by specific hydrolases after removal of the stimulus. However, linearly wired cascades are rare examples in biology, and multiplicity of enzymes that produce or degrade a second messenger often adds to complexity of these systems. Such a scenario is well described for eukaryotic cells which contain about 100 different isoforms of phosphodiesterases (PDEs) that hydrolyse cAMP or cGMP, respectively [2]. Similarly, c-di-GMP signalling in bacteria often comprises several dozens of c-di-GMP metabolizing enzymes, for example *Vibrio cholerae* possesses 53 diguanylate cyclases (DGCs) and c-di-GMP-specific PDEs [3], while *Escherichia coli*-K12 has 29 such enzymes

[4]. Multiplicity is also widespread at the level of effectors so that different types of sensor molecules can bind the same second messenger to elicit a cellular response. For example, in *E. coli* several c-di-GMP-binding proteins are described in the literature. Among these are the PilZ-domain proteins BcsA and YcgR [5], the BcsE protein characterized by a degenerate GGDEF domain [6], the MshEN-domain protein NfrB [7], the trigger PDE PdeR [8] and the poly-β-1,6-*N*-acetylglucosamine synthetases PgaCD [9,10].

Challenged by the question of how specificity can be achieved in networks comprising multiple second messenger synthases, hydrolases and effectors that are present at the same time in a cell, the concept of 'compartmentalized signaling' has been proposed [11–14] and proven to guide a broad spectrum of cellular responses to signals [15,16]. Locally acting signalosomes are often established by colocalization of synthases and/or hydrolases with a specific effector via protein-protein interactions. Such regulation has for example been reported for the biosynthesis of bacterial cellulose in *E. coli*, involving a specific DGC (DgcC) which associates with the synthase complex and acts as a local source for c-di-GMP by producing the molecule in the vicinity of BcsA. C-di-GMP binds to the PilZ domain of BcsA leading to stimulation of the glycosyltransferase domain [15,17]. Through localized activity, phosphodiesterases can also contribute to the establishment of 'nano-domains' of signalling molecules and serve as local sinks for the second messenger. This is particularly well studied in cardiomyocytes in which hotspots of cAMP accumulate only in areas of the cell that are devoid of PDEs and therefore thresholds of cAMP needed for activation of effectors are achieved in nano domains only [2,16,18].

In recent years, a growing number of structural studies of c-di-GMP effectors has revealed that c-di-GMP often acts as a 'protein matchmaker' and stabilizes protein-protein interactions or stimulates protein dimerization [10]. This concept is particularly evident in the antibiotic-producing soil bacteria of the genus *Streptomyces*, which are known to employ at least three different c-di-GMP effectors that all depend on the second messenger as stimulator of complex formation [19]. In these bacteria, c-di-GMP plays a central role in controlling developmental transitions between multicellular, filamentous vegetative hyphae and unicellular spores. During vegetative growth, the developmental master regulator BldD binds a tetrameric cage of c-di-GMP which induces dimerization of the protein and activates BldD as a repressor of sporulation genes [20]. In the early developmental stages, c-di-GMP also stabilizes the complex between a developmental sigma factor σ$^{WhiG}$ and the anti sigma factor RsiG, thus making the sigma factor inaccessible to RNA polymerase and preventing transcription of late sporulation genes [21]. Finally, during sporulation, a dimeric form of the glycogen debranching enzyme GlgX is activated through binding of c-di-GMP leading to degradation of the glucose storage compound glycogen, thus likely contributing to fitness of spores [22]. While several examples for local activation of a c-di-GMP effector system through the action of a specific DGC exist in the literature (see above and [15]), systems in which a PDE interrupts second messenger signalling by interfering with c-di-GMP-induced complex formation of effectors are rare and less well understood. In this study, we demonstrate using co-immunoprecipitation and co-elution assays that the PDE RmdB from *Streptomyces venezuelae,* containing both, a GGDEF and an EAL domain, interacts with the c-di-GMP effector σ$^{WhiG}$ via its GGDEF domain and propose that the PDE affects c-di-GMP signalling cascades via a global and a local mode of action.

## Results

### Deletion of *rmdB* leads to increased c-di-GMP in *S. venezuelae*

Synthesis of c-di-GMP from two molecules of GTP depends on the well-defined GGDEF domain, which is characteristic for all active DGCs. On the other hand, c-di-GMP-specific PDEs carry either an EAL or a HD-GYP domain [23,14]. *S. venezuelae* has 10 enzymes involved in c-di-GMP metabolism [19]. Within this set, five proteins have both GGDEF and EAL domains (CdgA, CdgC, CdgF, RmdA and RmdB) and two proteins have an HD-GYP domain (HdgA and HdgB) and thus all have the potential to act as PDEs. RmdA is an active PDE with residual DGC activity [24]. Analysis of CdgA in *S. coelicolor* suggests that the protein has DGC activity *in vivo* [25]. Similarly, CdgC has been shown to be an active DGC *in vitro* [26], while the function of CdgF still needs to be defined. The GYP residues in the HD-GYP domain of HdgB are missing, making it unlikely that the protein can hydrolyse c-di-GMP. Both, *hdgA* and *hdgB* are quite poorly expressed and deletion

PLOS Genetics

of these genes has no detectable consequences on growth or development when the strains are incubated on standard maltose-yeast extract-malt extract (MYM) medium [26].

The combined data indicate that RmdB is the central enzyme regulating c-di-GMP-mediated signalling cascades in *Streptomyces*. The protein has eight predicted transmembrane helices, a GGDEF and an EAL domain (Fig 1A). RmdB belongs to the most conserved c-di-GMP turnover enzymes in the *Streptomyces* pan genome and it is the most highly expressed of the c-di-GMP genes in *S. venezuelae* [26]. In a previous study, we demonstrated that RmdB is a functional PDE *in vitro*. Here, we examined the extent to which deletion of *rmdB* would affect the global pool of c-di-GMP *in vivo*. To address this question, we have extracted nucleotides from *S. venezuelae* wild type and the *rmdB* mutant grown for up to 20 hours in liquid MYM medium and quantified c-di-GMP using liquid chromatography-coupled tandem mass spectrometry (LC-MS/MS) in cells harvested every 2 hours after initial outgrowth phase for 8 h. As shown in Fig 1B, we detected increased c-di-GMP levels in the *rmdB* mutant in all analysed samples and conclude that RmdB acts as a PDE *in vivo* and balances the global c-di-GMP pool in *S. venezuelae* at all growth stages.

### The EAL motif and the GGDEF domain are crucial for the *in vivo* function of RmdB

In line with the model that increased levels of c-di-GMP in the *S. venezuelae rmdB* mutant stimulate dimerization of BldD and stabilizes the RsiG-$\sigma^{WhiG}$ complex to block cell differentiation, deletion of *rmdB* leads to a developmental delay. A

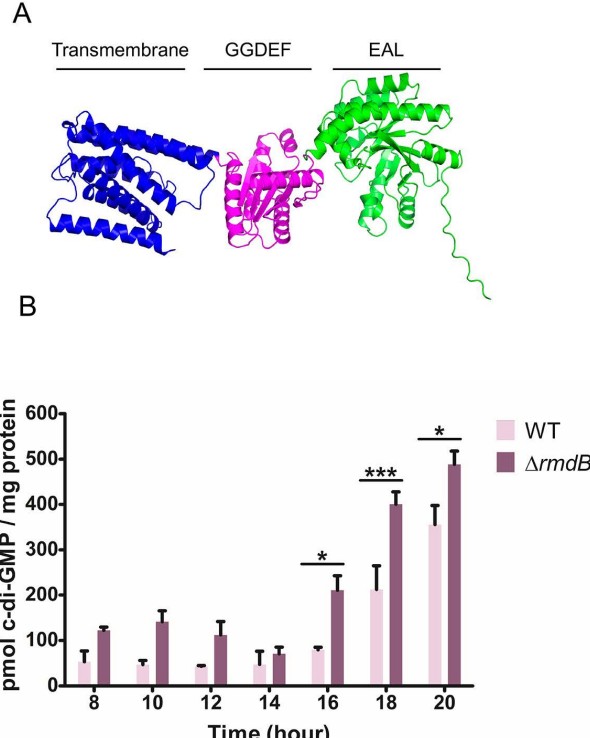

**Fig 1. Cellular c-di-GMP levels are increased in the *S. venezuelae rmdB* mutant throughout the life cycle.** (A) Domain architecture and structural predictions of RmdB. RmdB has predicted 8 transmembrane helices (amino acids 28-232; blue), a GGDEF domain (amino acids 258-421; pink) and an EAL domain (amino acids 426-670; green). The structural model was created with AlphaFold [27]. (B) Quantification of c-di-GMP in *S. venezuelae* cell extracts using LC-MS/MS. *S. venezuelae* wild type and Δ*rmdB* were harvested during late vegetative growth (8 to 10 h), transition to sporulation (12 to 16 h) and sporulation (18 to 20 h). Data are presented as the mean of biological replicates ± SD (n = 3). P values are calculated using a 2-way ANOVA test, *$p < 0.05$, ***$p < 0.001$.

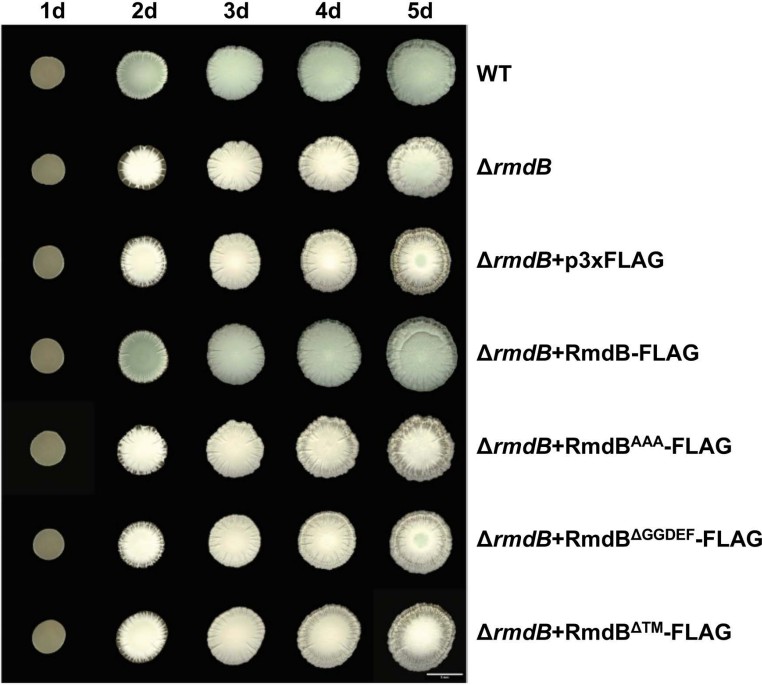

**Fig 2. Phenotypes of macrocolonies formed by *S. venezuelae* wild type, *rmdB* mutant and the *rmdB* mutant expressing different versions of *rmdB*-FLAG.** *S. venezuelae rmdB* mutant expressing either *rmdB*AAA-FLAG, *rmdB*ΔGGDEF-FLAG or *rmdB*ΔTM-FLAG from the Φ$_{BT1}$ phage integration site under the control of the native *rmdB* promoter were grown on MYM agar at 30°C for 5 days (d). *S. venezuelae* wild type (WT), the *rmdB* mutant and the mutant carrying the empty p3xFLAG vector were incubated under same conditions and served as controls. Images were taken daily using a stereomicroscope (Leica).

mature *S. venezuelae* colony becomes green after completion of the life cycle due to the biosynthesis of a polyketide pigment that is produced by spores only [28]. As shown in Fig 2 and in line with our previous studies [24,26], *S. venezuelae ΔrmdB* starts to become green only after prolonged incubation on maltose-yeast extract-malt extract (MYM) agar. We decided to make use of this phenotype for dissecting the physiological roles of individual protein domains by using complementation analysis. For that purpose, we have expressed four different FLAG-tagged gene versions from the integrative p3xFLAG vector [26] in the *S. venezuelae rmdB* mutant. These versions included the wild-type RmdB, the RmdBAAA variant in which the EAL motif was replaced by triple alanine (AAA) resulting in loss of PDE activity, a version in which the entire GGDEF domain (amino acids 258–421) was deleted, and a cytosolic version lacking the entire transmembrane region (amino acids 1–243). Consistent with our previous data, expression of wild-type *rmdB* restored the developmental defect of the mutant and this ability depends on the conserved EAL motif (Fig 2 and [24]). Interestingly, while the GGDEF motif *per se* is dispensable for RmdB function [24], the GGDEF domain must fulfil an important role, since an RmdB version lacking the GGDEF domain fails to complement the mutant phenotype. Of note, according to our western blot analysis, the RmdBΔGGDEF-FLAG version is present at similar levels as wild-type RmdB, supporting the conclusion that stability of the protein is not affected by deletion of the GGDEF domain (S1 Fig). In contrast, our data indicate that the transmembrane region contributes to protein stability in *S. venezuelae* since a cytosolic RmdB version (RmdBΔTM-FLAG) was neither detectable in the soluble nor in the pellet fraction of the cells (S1 Fig), despite the presence of the correct integrative plasmid in the relevant strain (our PCR results). Due to protein instability, we could not detect any effects of RmdBΔTM-FLAG on the phenotype of the *rmdB* mutant (Fig 2). In conclusion, our data indicate that the EAL motif and the GGDEF domain have crucial roles for RmdB function, while the transmembrane region contributes to protein stability in *S. venezuelae*.

## RmdB interacts with σ^WhiG through the GGDEF domain

Considering the key role of RmdB as a master PDE for the termination of c-di-GMP signalling cascades, we were wondering whether the protein is also involved in any local signalling modules and facilitates dissociation of effector complexes that are stabilized by c-di-GMP. In the σ^WhiG-(c-di-GMP)$_2$-RsiG complex, the bound c-di-GMP intercalated dimer is surface exposed and hence would be predicted to be accessible for a PDE (Fig 3A). To explore the possibility that RmdB interacts with σ^WhiG-(c-di-GMP)$_2$-RsiG, we first used co-purification after gene co-expression in *E. coli*. For that, we overexpressed the cytosolic version of *rmdB* tagged with Glutathione S-Transferase (GST) on the N-terminus from the pGEX vector and His-tagged *whiG* together with tag-less *rsiG* from the pIJ10914 vector [21] and used Ni-NTA matrix for protein purification. After elution and protein separation using SDS-PAGE, we found that GST-RmdB was eluted from Ni-NTA after co-expression with His-σ^WhiG and RsiG, but did not bind to Ni-NTA when expressed alone (Fig 3B). On the other hand, while His-σ^WhiG did not bind to glutathione sepharose, it could be purified from this matrix when GST-RmdB was present in the lysate (S5 Fig) Of note, when co-expressed with GST-RmdB, the amount of the anti sigma factor RsiG that was eluted from Ni-NTA was reduced, when compared to conditions in which RsiG was co-expressed with His-σ^WhiG only (Fig 3B). These results suggested that RmdB interacts with the σ^WhiG upon co-expression in *E. coli* and led us question whether this interaction also occurs in the native host, *S. venezuelae*. To address this possibility, we used co-immunoprecipitation and three different FLAG-tagged RmdB variants the expression of which was controlled by the native promoter in *S. venezuelae* Δ*rmdB* from the p3xFLAG vector [26]. We included FLAG-tagged wild-type RmdB, an enzymatically inactive RmdB^AAA version in which the EAL active site was mutagenized to AAA, a version without the GGDEF domain (RmdB^ΔGGDEF; see above and Fig 2) and the empty vector as our negative control. Cells were harvested and lysed after growth at 30°C for 18 hours. Total protein concentration of lysates was determined and equal amounts (20 µg) were used as input for the assay. The µMACS isolation kit (Miltenyi Biotec) was used to enrich FLAG-tagged RmdB variants on beads and both input samples and eluates were analysed by western blotting using specific antibodies against the FLAG-tag and σ^WhiG, respectively. Anti-FLAG detection confirmed that levels of RmdB-FLAG, RmdB^AAA-FLAG and RmdB^ΔGGDEF-FLAG were equal in the inputs, and they were pulled-down efficiently using the µMACS system at comparable amounts (Fig 3C). Strikingly, we detected a strong signal for co-purified σ^WhiG under both conditions, when lysates of strains expressing *rmdB*-FLAG or *rmdB*^AAA-FLAG, respectively, were used. Since both enzymatically active and inactive RmdB efficiently co-immunoprecipitated with σ^WhiG, we conclude that the two proteins interact *in vivo* regardless of the enzymatic activity of the PDE. Notably, when lysates of the strain expressing *rmdB*^ΔGGDEF-FLAG were used, the amount of co-eluted σ^WhiG was significantly reduced when compared to samples containing *rmdB*-FLAG or *rmdB*^AAA-FLAG. Importantly, both samples contained the same amounts of input RmdB (Fig 3C and 3D) and AlphaFold3 [29] predicts that RmdB^ΔGGDEF structure closely resembles that of full-length RmdB (S7 Fig) This observation suggests that the GGDEF domain mediates interaction between RmdB and σ^WhiG. To verify this conclusion, we co-expressed His-σ^WhiG and GST-tagged GGDEF domain of RmdB in *E. coli* BL21(pLysS) and used Ni-NTA for protein purification. GST-GGDEF$_{RmdB}$ could be co-purified with His-σ^WhiG (S6 Fig) supporting our finding that RmdB and His-σ^WhiG interact via the GGDEF domain. Noteworthy, RsiG was not detected in any of the eluates (S4 Fig), supporting the model that interaction between RmdB and σ^WhiG either results in release of RsiG from the σ^WhiG-(c-di-GMP)$_2$-RsiG complex or prevents its formation under specific conditions.

## Bacterial Adenylate Cyclase Two-Hybrid (BACTH) System identifies RmdB as a central hub for interactions with DGCs

c-di-GMP signalling in bacteria frequently involves interactions between antagonistically acting PDEs and DGCs [15,30]. Having found that RmdB interacts with σ^WhiG, we next addressed the question of whether a 'partner DGC' exists for RmdB using the BACTH system [31]. In particular, we used BACTH to investigate *in vivo* interactions between RmdB and all GGDEF-, EAL-, HD-GYP-domain proteins produced by *S. venezuelae* in the *cya* deficient *E. coli* BTH101 reporter strain. For these assays, the proteins of interest were genetically fused to two complementary fragments, T25 and T18, that

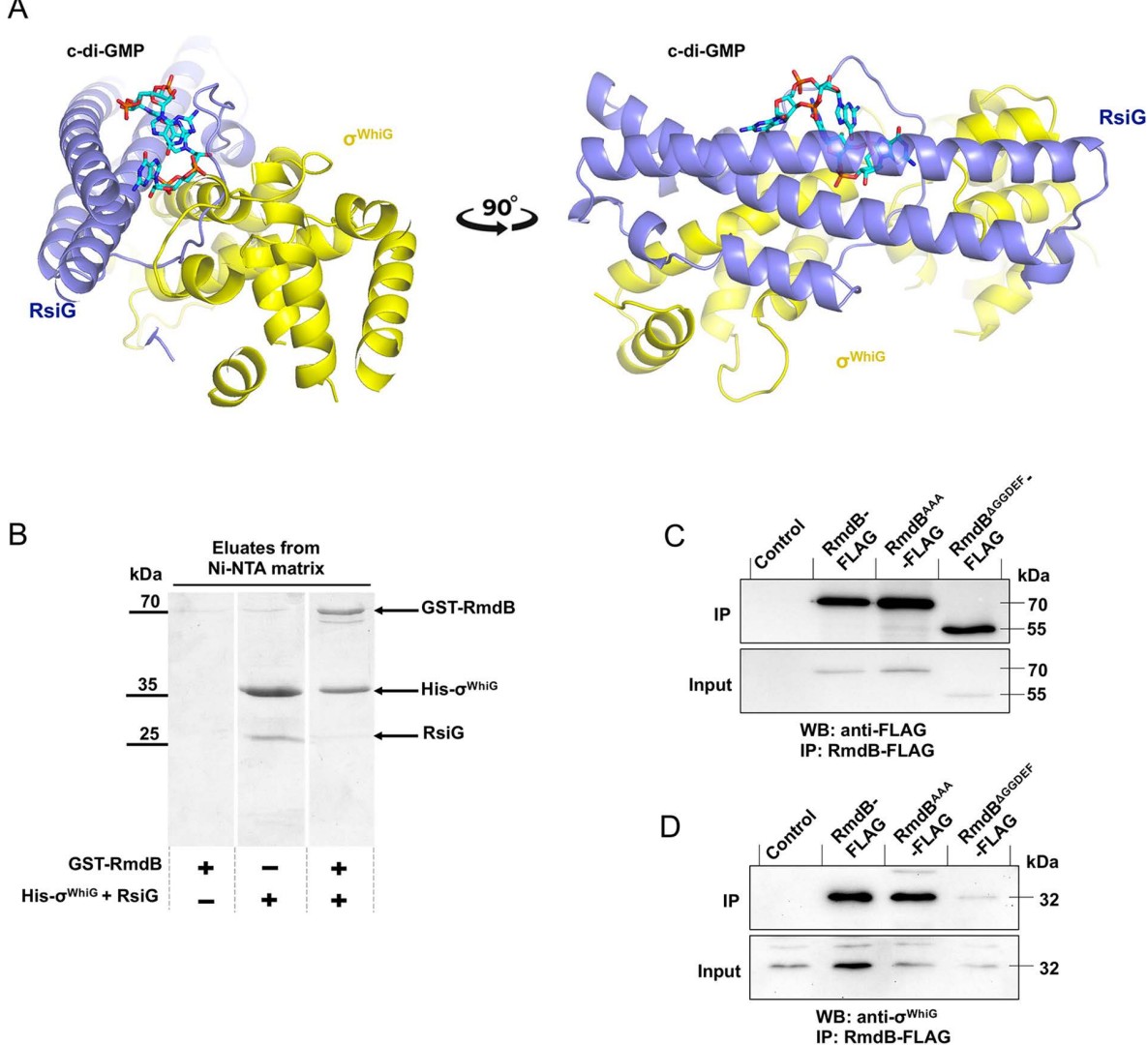

**Fig 3. RmdB interacts with σ<sup>WhiG</sup>.** (A) Ribbon diagram of the overall structure of the *S. venezuelae* RsiG-(c-di-GMP)$_2$-σ$^{WhiG}$ complex [21]. The anti sigma factor RsiG is colored in blue and σ$^{WhiG}$ in yellow. The two partially intercalated c-di-GMP molecules are shown as sticks. (B) Co-purification of GST-RmdB and His- σ$^{WhiG}$ from *E. coli* BL21 pLysS Rosetta. GST-tagged *rmdB* was expressed from the pGEX.6p1 vector. His-*whiG* and tag less *rsiG* were expressed from pCOLADuet-1 from two separate multiple cloning sites [21]. Gene expression was induced with 250 µM IPTG and cells were incubated in LB at 16°C overnight. Purification of GST-RmdB, His-WhiG and RsiG was performed using the Ni-NTA matrix. The eluates were analysed via SDS-PAGE. (C, D) Co-immunoprecipitation (Co-IP) analysis using RmdB-FLAG, RmdB$^{AAA}$-FLAG and RmdB$^{ΔGGDEF}$-FLAG in *S. venezuelae*. *rmdB*-FLAG, *rmdB*$^{AAA}$-FLAG and *rmdB*$^{ΔGGDEF}$-FLAG were expressed in the *S. venezuelae rmdB* mutant. Cells were grown in MYM at 30°C and 180 rpm for 18 hours. FLAG-tagged proteins were immunoprecipitated using FLAG-tag-specific magnetic beads (Miltenyi Biotec). *S. venezuelae* Δ*rmdB* carrying the empty p3xFLAG plasmid served as a negative control. After Co-IP, eluates (=IP) and cell lysates (=Input) were analysed using western blotting (WB) and either the monoclonal anti-FLAG (Sigma) (C) or the polyclonal anti-WhiG (D) antibodies [21]. 20 µg total protein was used as input for each sample. Panels B, C and D show representative images of 3 independent experiments.

constitute the catalytic domain of *Bordetella pertussis* adenylate cyclase. Interactions between the relevant fusion proteins results in functional complementation between the T25 and T18 fragments and cAMP production. cAMP then binds the catabolite activator protein CAP to trigger expression of the lactose or maltose operons, yielding a characteristic phenotype on indicator medium [31]. We fused the relevant adenylate cyclase domain to the C-terminal end of RmdB

to minimize the risk of interference with protein integration into the membrane but found the resulting fusion proteins to be unstable in *E. coli* (our western blot data). Therefore, we excluded the regions encompassing the RmdB membrane domain during plasmid design and instead used fusion proteins representing the cytosolic version of RmdB (Δ1–238aa) either carrying T18 (on pUT18C) or T25 (from pKT25) in subsequent analyses. Strains expressing fusions of both adenylate cyclase domains to the leucine zipper domain of GCN4 (zip) served as a positive control, whereas strains carrying the two empty vectors, pKT25 and pUT18, functioned as negative controls. Our analysis revealed that T18-RmdB$^{Δ1-238aa}$ and T25-RmdB$^{Δ1-238aa}$ interact, indicating that RmdB can form dimers or oligomers (Figs 4, S8A, and S8B). Similarly, we observed homomeric interactions for CdgA, CdgB, CdgC, CdgE and HdgB. Importantly, our data also revealed that T18-RmdB$^{Δ1-238aa}$ interacts with CdgA, CdgB, CdgC and CdgE suggesting that RmdB serves as an interaction hub with the ability to make contacts to four different DGCs. Of note, we did not detect interactions between T18-RmdB$^{Δ1-238aa}$ or T25-RmdB$^{Δ1-238aa}$ with σ$^{WhiG}$ in BACTH assays, which may be due to suboptimal folding of the fusion proteins or the requirement for a tripartite RmdB-σ$^{WhiG}$-RsiG system. However, these data raised the question as to whether any of the DGCs that interact with RmdB might also interact with σ$^{WhiG}$. Addressing this possibility, we found that out of the four relevant DGCs, only CdgE-T18 interacts with T25-σ$^{WhiG}$ (Figs 4, S8C, and S8D). Collectively, our co-immunoprecipitation and BACTH data

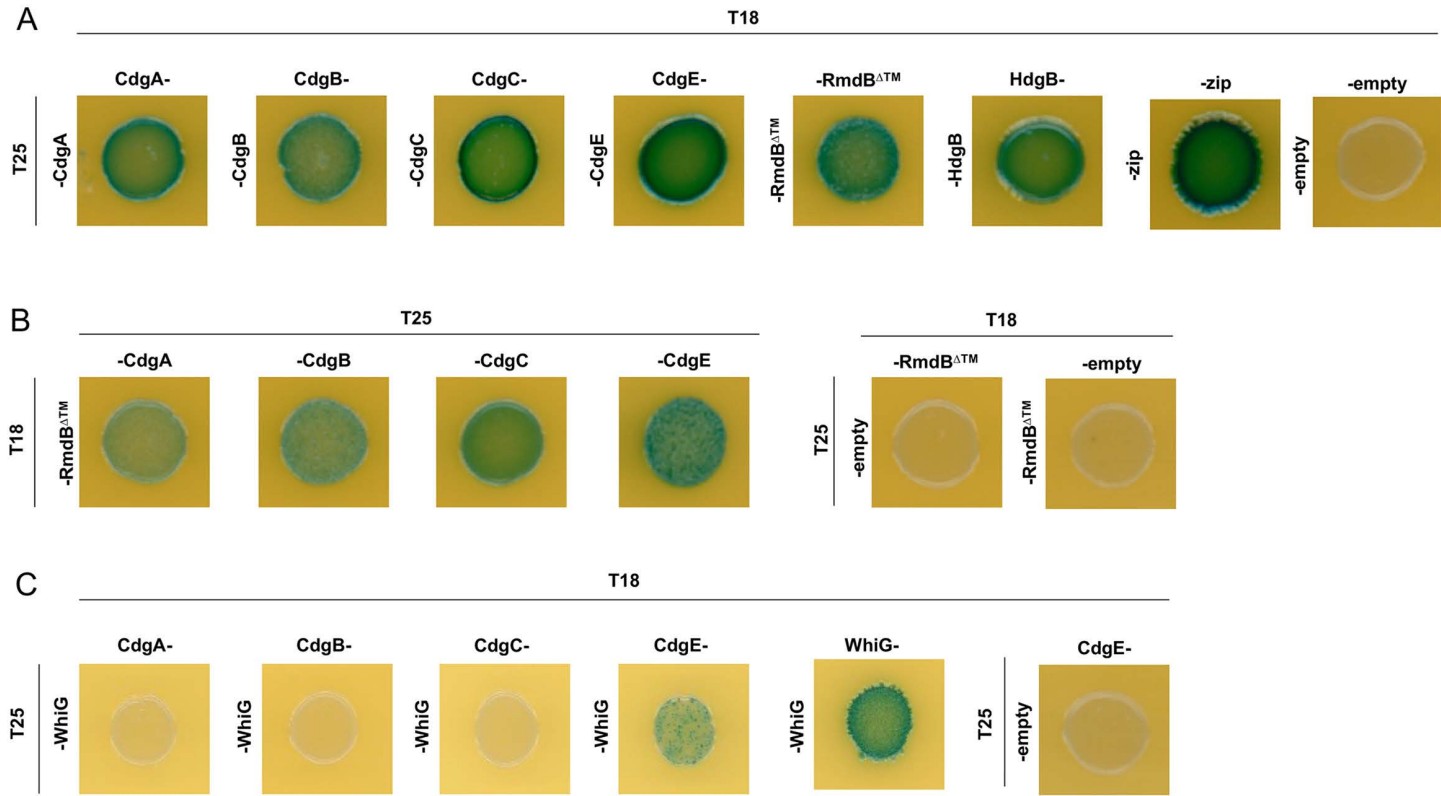

**Fig 4. RmdB interacts with multiple DGCs in BACTH assays.** pKT25 and pUT18/pUT18C plasmids containing the indicated genes were co-transformed into *E. coli* BTH101. Several clones carrying both plasmids were combined in 1 ml sterile 1x PBS and 10 µl were spotted on LB agar supplemented with 5-Bromo-4chloro-3indolyl-ß-D-galactopyranoside (X-gal) 60 µg/ml, 0.5 mM IPTG, ampicillin (100 µg/µl) and kanamycin (50 µg/µl). Plates were incubated for either 40 hours at 30°C **(A, B)** or for 56 hours at 25°C **(C)**. Strains expressing fusions of both adenylate cyclase domains to the leucine zipper domain of GCN4 (zip) served as positive control. Empty vectors combinations were used as negative controls. Cytosolic fraction of RmdB (Δ1–238aa) was used in BACTH assays. The entire screening plates showing additional controls are depicted in S8 Fig.

suggest that, the DGCs CdgA, CdgB, CdgC and CdgE can interact with RmdB, and that CdgE, RmdB and σ<sup>WhiG</sup> constitute a DGC-PDE-effector local signalling module (Fig 5) controlling *Streptomyces* development.

## Discussion

Second messenger signaling systems provide bacteria with an exceptional capacity for rapid and adjusted responses to environmental cues. In *Streptomyces*, the second messenger, c-di-GMP, functions as the central regulator of the developmental transition from vegetative hyphae to sporulation through the modulation of key effectors, including the transcriptional repressor BldD and the sigma factor σ<sup>WhiG</sup> [19–21]. While mechanisms for activation of these complexes by c-di-GMP are structurally and functionally well-characterized, the molecular events that underpin how the c-di-GMP signal is controlled in a timely manner to affect their activities is less well understood. Here, we identify the phosphodiesterase RmdB as a dual-function regulator that globally depletes c-di-GMP and locally interacts with a c-di-GMP effector complex to achieve signal termination. Our findings support a model in which RmdB functions as an anti sigma factor antagonist that competes with the anti sigma factor RsiG for binding to its partner protein σ<sup>WhiG</sup> - a previously unrecognized role for a phosphodiesterase in bacteria.

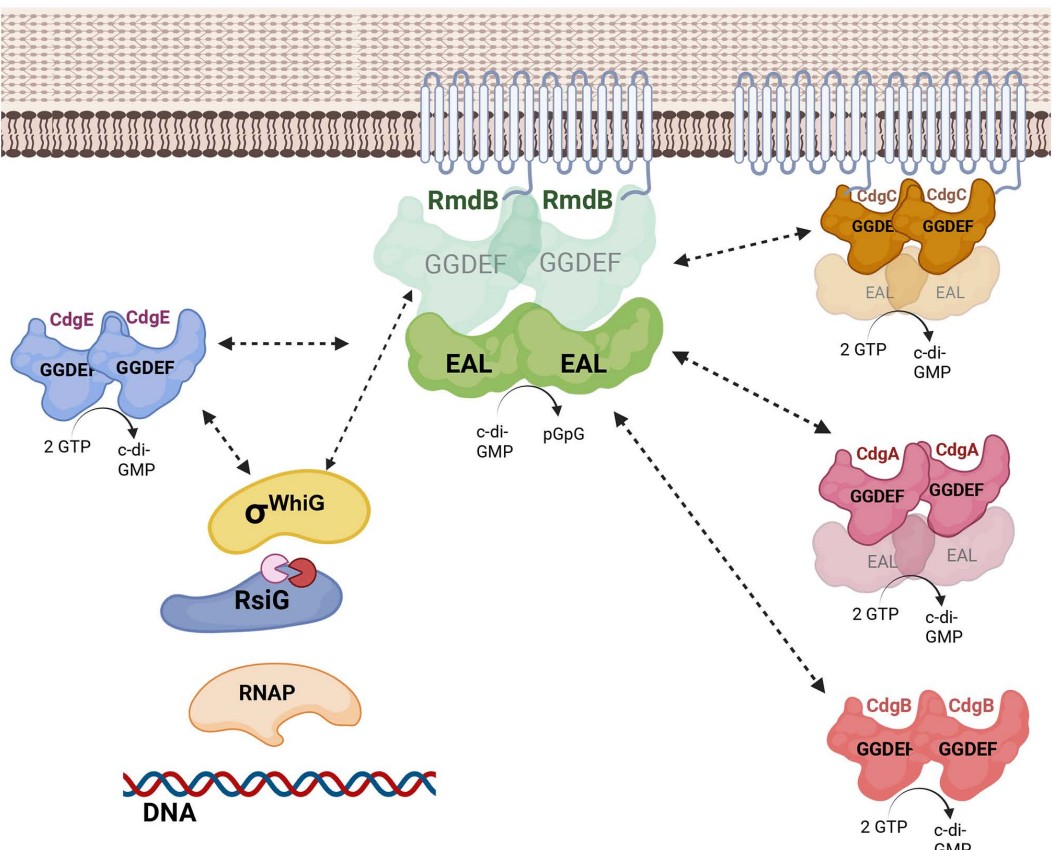

**Fig 5. Schematic illustration of RmdB as a hub for protein-protein interactions in the *Streptomyces* c-di-GMP signalling network.** Our BACTH assays revealed that the PDE RmdB can form homodimers and interact with the DGCs CdgA, CdgB, CdgC, and CdgE. CdgE also interacts with σ<sup>WhiG</sup> in *E. coli*. Our results obtained using co-immunoprecipitation demonstrate that σ<sup>WhiG</sup> forms contacts with the enzymatically inactive GGDEF domain of RmdB, supporting the conclusion that RmdB acts in a local manner to antagonize binding of the anti sigma factor RsiG to σ<sup>WhiG</sup>. Enzymatically inactive protein domains are shown in fade colours. RsiG binds 2 molecules of c-di-GMP shown as little pac-mans. RNAP: RNA polymerase. Created in BioRender. Tschowri, N. (2026) https://BioRender.com/dzyzlc3.

RmdB is a tandem protein carrying conserved GGDEF and EAL domains. Several lines of evidence described in our previous studies and in this work clearly demonstrate that RmdB acts as a PDE via the EAL domain and that this activity is central for its *in vivo* function. RmdB hydrolyzed c-di-GMP to the linear pGpG molecule in biochemical assays and a strain expressing a mutagenized RmdB version with triple alanines (AAA) in place of the EAL residues from the native chromosomal locus had a developmental delay, exactly as the *rmdB* null mutant [24]. Here, we confirm the important role of the EAL active site for protein function and show that a mutated RmdB version carrying AAA residues instead of EAL cannot complement the mutant phenotype, when expressed from an integrative vector and controlled by the native promoter (Fig 2). Importantly, our quantification of cellular c-di-GMP levels (Fig 1B) revealed that RmdB represents an important c-di-GMP sink by keeping global c-di-GMP at lower levels at all developmental stages and thus ensuring controlled progression through the life cycle.

Although RmdB contains a conserved GGDEF motif, all our attempts to demonstrate DGC enzymatic activity for this domain have failed. Incubation of RmdB with GTP did not result in any reaction products and an allele carrying the mutagenized GGAAF motif in the GGDEF site fully complemented the differentiation defect of the *rmdB* mutant [24]. However, while the enzymatic active site, i.e., the GGDEF motif is dispensable for protein function, the GGDEF domain is not, as an RmdB version lacking the entire GGDEF domain (amino acids 258–421) cannot complement the mutant phenotype despite protein stability and predicted proper folding (Figs 2, S1 and S7). Strikingly, our data explain the importance of this domain, as it serves as an interaction domain with $\sigma^{WhiG}$. Our co-immunoprecipitation analysis using *S. venezuelae* showed that the amount of enriched $\sigma^{WhiG}$ is strongly reduced when RmdB$^{\Delta GGDEF}$-FLAG is used as a bait (compare lane 3 and 4 in Fig 3D) and co-elution experiments from *E. coli* demonstrate that GST-GGDEF$_{RmdB}$ co-purifies with His-$\sigma^{WhiG}$ (S6 Fig). Interestingly, the GGDEF domain of the phosphodiesterase PdeB from *Shewanella putrefaciens* also mediates interaction between PdeB and the polar landmark protein HubP, which is essential for polar localization and full activity of the enzyme [32]. Thus, our finding that the GGDEF domain of RmdB is central for binding to $\sigma^{WhiG}$ supports the concept that GGDEF domains can evolve to serve as protein-protein interaction platforms.

Our BACTH screen identifies RmdB as a promiscuously interacting protein with the ability to make contacts to the DGCs CdgA, CdgB, CdgC and CdgE (Figs 4, 5 and S8), raising the possibility that these interactions may conditionally modulate the outcome of the developmental switch controlled by BldD$_2$(c-di-GMP)$_4$ and $\sigma^{WhiG}$-(c-di-GMP)$_2$-RsiG. *E. coli in vivo* two-hybrid approaches are based on fusion of proteins which can affect protein folding with consequences on their stability and/or activity. Therefore, generation of a certain fraction of false positives or failures in capturing all interactions cannot be excluded using these approaches. However, the observation that a 'master PDE' serves as an interaction hub to control a cellular decision as revealed in this work, is not limited to *Streptomyces*. In *E. coli*, PdeR which has a PAS-GGDEF-EAL domain architecture, makes contacts to four different DGCs and serves as a central decision-making component of the CsgD-dependent biofilm matrix biosynthesis switch [30].

What is the physiological relevance of the RmdB-$\sigma^{WhiG}$ interaction? Our quantification of c-di-GMP levels across *S. venezuelae* development revealed that they range from 10-75 pmol per mg total protein, with a strong decline during the transition to sporulation [22]. This would correspond to c-di-GMP concentrations of 1-7.5 µM, if the protein concentration of the cytoplasm were ~100 µg/µl [19]. Considering that [RsiG + $\sigma^{WhiG}$] binds c-di-GMP with a relatively high affinity of 0.4 µM [21] it seems likely that a global drop of c-di-GMP to ~ 1 µM would not be sufficient to cause full dissociation of the complex and therefore RmdB is needed to facilitate the release of the sigma factor through its local action. In line with this model, the amount of RsiG that co-elutes with His-$\sigma^{WhiG}$ is reduced after co-expression of the sigma-anti sigma together with RmdB in *E. coli* (Fig 3B). However, our westernblot data obtained using co-immunoprecipitation and FLAG-tagged RmdB variants expressed in *S. venezuelae* suggest that the enzymatic activity of RmdB is not crucial for the prevention of RsiG binding to $\sigma^{WhiG}$, as we failed to detect co-immunoprecipitated RsiG under both conditions, when using either the enzymatically active or inactive version of RmdB (Figs 3C, 3D and S4). While we cannot exclude that failures of co-eluted RsiG detection are due to sensitivity limits of our approach, we conclude that protein-protein interaction between RmdB

and σ<sup>WhiG</sup> is central for the local action of RmdB as antagonist of an anti sigma factor that promotes displacement of RsiG. The isolated EAL domain of RmdB is enzymatically fully active [24], still an RmdB version missing the GGDEF domain that mediates interaction between RmdB and σ<sup>WhiG</sup> fails to complement the developmental defect of the *rmdB* mutant. These data suggest that interaction between RmdB and σ<sup>WhiG</sup> are important for the developmental program progression, likely due to RmdB-mediated release of σ<sup>WhiG</sup> from RsiG. However, understanding how RmdB supports RNA-polymerase-σ<sup>WhiG</sup> holo enzyme formation remains a challenge that we will address in our future work.

## Materials and methods

### Bacterial strains and growth conditions

Bacterial strains used in this study are listed in S1 Table. For standard cultivation, *E. coli* strains were grown in LB at 37 °C and *S. venezuelae* strains in liquid or solid MYM media supplemented with trace element solution at 30 °C. Liquid cultures were inoculated with spores to a final concentration of $10^6$ CFU/ml. To track *S. venezuelae* development on solid media, 12 µl of $2 \times 10^5$ CFU/µl spores were spotted on MYM and incubated at 30 °C. The resulting macrocolonies were imaged daily using an S9 i stereomicroscope (Leica). When selection was required, media was supplemented with 50 µg/ml kanamycin, 50 µg/ml apramycin, 15 µg/ml chloramphenicol or 50 µg/ml hygromycin.

### Construction of plasmids

All strains, plasmids, and oligonucleotides used in this study are listed in S1 Table. Plasmids used for BACTH assay were designed using the restriction and ligation approach. Genes of interest were amplified from genomic *S. venezuelae* DNA with primers listed in S1 Table. For cloning of *cdgA*, *cdgB*, *cdgC*, *cdgD*, *cdgE* and *hdgA* into pUT18 and pKT25 vectors, XbaI and KpnI were used. HindIII and EcoRI served for cloning of *cdgF* into pKNT25 and of *rmdA* into pUT18. *rmdA* was cloned into pKT25 using EcoRI and XbaI. *rmdB* was amplified and cloned into XbaI/KpnI-digested BACTH plasmids (S1 Table). Site-directed mutagenesis was performed to delete the transmembrane region of RmdB (1–238 aa). pKT25-*rmdB* and pUT18C-*rmdB* plasmids were used as templates, and the transmembrane helices (1–238 aa) were deleted via backbone PCR using PRBC15_Fr/PRBC15_Rv and PRBC15_Fr/PRBC16_Rv primer pairs, respectively (S1 Table). The amplified fragment was phosphorylated and ligated. The resulting plasmids (pEBC9 and pECBC10) were used for BACTH assays. For complementation analysis, plasmids were generated via site-directed mutagenesis or Gibson assembly using primers listed in S1 Table. The p3xFLAG-*rmdB* (pSVJH02) plasmid [26] was used as a template to perform deletions of the transmembrane helices (1–243 amino acids), the GGDEF domain (28–421 amino acids), and for mutagenesis of the EAL motif to AAA to generate p3xFLAG-*rmdB*<sup>ΔTM</sup>, p3xFLAG-*rmdB*<sup>ΔGGDEF</sup> and p3xFLAG-*rmdB*<sup>AAA</sup> plasmids, respectively. p3xFLAG plasmid was cut with NdeI and AvrII for Gibson assembly reactions. For co-purification of WhiG and GST-RmdB<sub>GGDEF</sub>, GST-RmdB<sub>GGDEF</sub> was amplified using 15b_GSTGGDEF_Nco_Fwd and 15b_GSTGGDEF_Bgl_Rev primers and cloned into digested pET15b.

### Construction of *S. venezuelae* strains expressing *rmdB* variants for complementation of the *rmdB* mutant

For complementation analysis, the p3xFLAG-*rmdB*<sup>ΔTM</sup>, p3xFLAG-*rmdB*<sup>ΔGGDEF</sup> and p3xFLAG-*rmdB*<sup>AAA</sup> plasmids carrying the genes under the control of the native *rmdB* promoter were conjugated into *S. venezuelae* Δ*rmdB* strain as described in [24]. p3xFLAG vector integrates at the ΦBT1 phage site on the chromosome.

### BACTH assays

Bacterial Adenylate Cyclase-Based Two-Hybrid (BACTH) assays were performed according to [31]. Relevant plasmids for each tested combination were co-transformed in chemo-competent *cya* deficient *E. coli* BTH101 cells. Heat shock was performed for 90 s at 42 °C. Cells were recovered in 1 ml LB for 1 h at 37 °C and 180 rpm. Co-transformants were selected

on LB-agar containing 60 µg/ml 5-bromo-4-chloro-3-indolyl-β-d-galactopyranoside (ß-X-Gal), 100 µg/ml ampicillin, 50 µg/mL kanamycin, and 0.5 mM Isopropyl β-d-1-thiogalactopyranoside (IPTG). Plates were incubated for 48 h at 30 °C. For protein-protein interaction analyses, 3 clones containing the relevant plasmid combination were pooled in 1 ml 1x PBS. 10 µl of this mixture were spotted on LB plates containing 60 µg/ml ß-X-Gal, 100 µg/ml ampicillin, 50 µg/ml kanamycin and 0.5 mM IPTG. Plates were incubated at 30 °C or 25 °C for either 40 or 56 hours before imaging.

## Co-purification of RmdB and σ$^{WhiG}$ from *E. coli*

5 ml LB containing relevant antibiotics were inoculated with *E. coli* BL21 pLysS Rosetta carrying pIJ10914, pECJH14 or both plasmids in combination (S1 Table), and incubated at 30°C overnight. The next day, 500 ml LB supplemented with either 50 µg/mL kanamycin, 100 µg/mL ampicillin or both were inoculated with 1:100 dilution of the overnight culture with subsequent cultivation at 30 °C until an $OD_{600}$ of 0.5. Gene expression was induced using 250 µM IPTG and cells were grown at 16 °C overnight. Next day, cells were harvested at 4°C, 7000xg for 20 minutes. Pellets were resuspended in 3 ml cold 1x PBS containing proteinase inhibitor (Roche) and cells were disrupted using sonication (40% amplitude, 2 min of duration, 3 microns). The suspension was centrifuged at 17000 rpm for 40 min. Total protein concentration was determined via Bradford assay (Roth) and adjusted to 60 mg/ml. 150 µl Ni-NTA agarose was placed in a 10 ml column. The matrix was washed with 10 ml His-Lysis buffer (50 mM Tris pH 7.5, 300 mM NaCl, 20 mM imidazole, 10% glycerol, 0.1% Triton X-100, 1 mM β-mercaptoethanol). 500 µl of the cell lysate was added to the column together with 1.5 ml of His-Lysis buffer and the mixture was incubated at 4 °C for 1 hour while rolling. The column was washed with 10 ml of His-Washing buffer (50 mM Tris pH 8, 300 mM NaCl, 50 mM imidazole, 10% glycerol, 0.1% Triton X-100, 1 mM β-mercaptoethanol). The proteins were eluted 3 times with 150 µl His-Elution buffer (50 mM Tris pH 7.5, 300 mM NaCl, 250 mM imidazole, 10% glycerol, 0.1% Triton X-100, 1 mM β-mercaptoethanol) and analysed using SDS-PAGE.

## µMACS epitope tag protein isolation in *S. venezuelae*

Co-immunoprecipitation experiments using FLAG-tagged RmdB as bait were performed using *S. venezuelae* Δ*rmdB* strains expressing *rmdB*-FLAG, *rmdB*$^{AAA}$-FLAG and *rmdB*$^{ΔGGDEF}$-FLAG controlled by the native *rmdB* promoter. *S. venezuelae* Δ*rmdB* carrying the empty p3xFLAG plasmid served as a negative control. 50 ml MYM + TE supplemented with 50 µg/mL hygromycin were inoculated with respective *S. venezuelae* spores (final concentration 1x10$^6$ CFU/ml) and incubated at 30 °C and 180 rpm for 18 hours. Culture was centrifuged at 4 °C at 4500 rpm for 10 minutes. The pellet was washed three times with 25 ml of ice-cold 1x PBS (pH 7.4). After washing, the bacterial pellet was suspended in 2 ml lysis buffer (20 mM Tris-HCl pH 8, 5 mM EDTA pH 8) containing protease inhibitors (Roche). Bacteria were lysed via sonication (12 x 15 sec on/ 30 sec off, 70%, 4 micro-tips), then centrifuged at 4 °C, 10000 rpm for 15 minutes. The protein concentration was determined via Bradford assay (Roth). Co-immunoprecipitation was performed using Miltenyi Biotec's µMACS DYKDDDDK isolation kit (130-101-591). 50 µl anti-tag microbeads were added to 1 ml of cell lysate and incubated for 2 hours on a rotating wheel at 4°C. The µ column was placed in the magnetic field and equilibrated with 200 µl of lysis buffer. The cell lysate was applied to the column, and the flow-through was collected. The column was rinsed 4 times with 200 µl of lysis buffer, then once with 100 µl wash buffer (20 mM Tris-HCl pH 7.5). For elution, 20 µl of pre-heated 95 °C elution buffer (50 mM Tris-HCl pH 6.8, 50 mM DTT, 1% SDS, 1 mM EDTA, 0.005% bromophenol blue, 10% glycerol) was applied to the column for 5 minutes at room temperature. Finally, the proteins were eluted two-times using 50 µl of pre-heated elution buffer.

## Western blotting

For the detection of FLAG-tagged RmdB and WhiG, eluates acquired using µMACS epitope tag protein isolation from *S. venezuelae* were used. Input samples were diluted to 1 µg total protein/µl in loading buffer, boiled for 5 minutes and centrifuged at 13000 rpm for 5 minutes. 20 µg of each input sample and eluate were separated on a 15% SDS-PAGE gel.

Following the electrophoresis, proteins were transferred to a polyvinylidene difluoride (PVDF, Roth) membrane by electroblotting using a semi-dry blotting system from Bio-Rad. For detection of FLAG-tagged proteins, the anti-FLAG antibody (Sigma) and the HRP-conjugated anti-mouse (Roth) were used at 1:10000 and 1:2000 dilutions, respectively. For detection of WhiG, a polyclonal anti-WhiG antibody raised in rabbit [21] and the HRP-conjugated anti-rabbit (Cytiva) were used at 1:10000 dilutions. Antibodies were added to the membrane iphosphate-buffered saline pH 7.4, 0.1% (v/v) Tween 20 (PBST) buffer containing 5% (v/v) non-fat dry milk (skim milk). Blots were developed using Clarity Western ECL Substrate (BioRad) and a ChemoStar ECL Imager (Intas Pharmaceuticals Limited).

### Nucleotide extraction from *S. venezuelae*

*S. venezuelae* strains were cultivated in 100 ml MYM media. All experiments were performed as biological triplicates. Samples were taken every 2 hours in the period from 8 to 20 hours after the start of cultivation. 5 ml of cell culture was taken for subsequent nucleotide extraction, and 2x 1 ml samples of cell culture were taken for protein determination. Nucleotide extraction was performed as described in [33] and adapted to the requirements of *Streptomyces*. All samples for nucleotide extraction were resuspended in 800 µl Extraction Mixture II (Methanol/ Acetonitrile/ Water (2:2:1)) and transferred to screw-cap tubes with pre-placed zirconia beads (Lysing Matrix B, MP Biomedicals). Prior to the homogenization step, the samples were placed in liquid nitrogen for 15 seconds and then heated at 95 °C for 10 minutes. Homogenization was performed by vigorously shaking the zirconia beads in a FastPrep-24 5G apparatus (MP Biomedicals). Five cycles of 30 seconds of shaking (6 m/s) and 1 minute of rest at 4 °C were performed. The cell suspension was centrifuged for 15 minutes at 4 °C (max. rpm), and the supernatant was transferred to 2 ml reaction tubes. The remaining cell pellets were resuspended in 600 µl Extraction Mixture I (Methanol/ Acetonitrile (1:1)), and homogenization as well as extraction in 600 µl Extraction Mixture I was repeated as described above. The collected supernatants (approx. 2 ml) were combined and stored overnight at -20°C. Precipitated proteins were removed by centrifugation, and the supernatant was transferred to a new 2 ml reaction tube. This precipitation step was repeated twice. Samples were then dried at 45°C using a SpeedVac Plus SC110A apparatus (Savant) and analyzed by LC-MS/MS as described in [33]. Samples for protein determination were resuspended in 800 µl of 0.1 M NaOH and transferred to 2 ml screw-cap tubes containing zirconia beads (0.1 mm silica beads, Biozym) for heating to 98°C for 10 minutes. After cooling on ice, homogenization was performed by vigorously shaking in a FastPrep-24 5G apparatus (MP Biomedicals). Five cycles of 30 seconds of shaking (6 m/s) and 1 minute of rest at 4 °C were performed. The cell lysates were centrifuged for 15 minutes at 4 °C and max. rpm. The supernatants were removed, and the homogenization step was repeated. Supernatants were combined and the total protein concentration was determined using the Bradford assay. The following formula was used to normalize the nucleotide concentration to the total protein concentration:

$$\frac{X\ [nM].\ 200\ \mu L}{cV\ [mL].\ c590\ [\frac{\mu g}{ml}]} = \frac{X\ [pmol]}{Protein\ [mg]}.$$

### Supporting Information

**S1 Fig. Western Blot analysis to study protein levels of different RmdB variants.** *S. venezuelae rmdB* mutant carrying either *p3xFLAG-rmdB*, *p3xFLAG-rmdB*$^{\Delta GGDEF}$, *p3xFLAG-rmdB*$^{AAA}$, or *p3xFLAG-rmdB*$^{\Delta TM}$ was grown in MYM complemented with hygromycin (50 µg/ml) at 30°C and 180 rpm. Samples were taken after 12, 16 and 20 hours of growth. 20 µg total protein were used for each sample. The anti-FLAG antibody (Sigma) was used for detection.
(DOCX)

**S2 Fig. Co-purification of GST-RmdB and His- σ$^{WhiG}$ from *E. coli* BL21 pLysS Rosetta (non-processed version of the image shown in Fig 3B).** GST-tagged *rmdB* was expressed from the pGEX.6p1 vector. His-*whiG* and tag less *rsiG* were expressed from pCOLADuet-1 from two separate multiple cloning sites [21]. Gene expression was induced with

250 µM IPTG and cells were incubated in LB at 16°C overnight. Purification of GST-RmdB, His-WhiG and RsiG was performed using the Ni-NTA matrix. The eluates were analysed via SDS-PAGE.
(DOCX)

**S3 Fig. Co-immunoprecipitation (Co-IP) analysis using RmdB-FLAG, RmdB^AAA^-FLAG and RmdB^ΔGGDEF^-FLAG in *S. venezuelae* (non-processed images from Fig 3C (A) and 3D (B)): *rmdB*-FLAG, *rmdB*^AAA^-FLAG and *rmdB*^ΔGGDEF^-FLAG were expressed in the *S. venezuelae rmdB* mutant.** Cells were grown in MYM at 30°C and 180 rpm for 18 hours. FLAG-tagged proteins were immunoprecipitated using FLAG-tag-specific magnetic beads (Miltenyi Biotec). *S. venezuelae* ΔrmdB carrying the empty p3xFLAG plasmid served as a negative control. After Co-IP, eluates (=IP) and cell lysates (=Input) were analysed using western blotting (WB) and either the monoclonal anti-FLAG (Sigma) (C) or the polyclonal anti-WhiG (D) antibodies [33]. 20 µg total protein was used as input for each sample.
(DOCX)

**S4 Fig. Co-immunoprecipitation (Co-IP) analysis failed to detect interaction between RmdB and RsiG. *rmdB*-FLAG, *rmdB*^AAA^-FLAG and *rmdB*^ΔGGDEF^-FLAG were expressed in the *S. venezuelae rmdB* mutant.** Cells were grown in MYM at 30°C and 180 rpm for 18 hours. FLAG-tagged proteins were immunoprecipitated using FLAG-tag-specific magnetic beads (Miltenyi Biotec). *S. venezuelae* ΔrmdB carrying the empty p3xFLAG plasmid served as a negative control. After Co-IP, eluates (=IP) and cell lysates (=Input) were analysed using western blotting (WB) and the polyclonal anti-RsiG antibody [33]. 20 µg total protein were used as input for each sample.
(DOCX)

**S5 Fig. Co-purification of GST-RmdB and His- σ^WhiG^ from *E. coli* BL21 pLysS Rosetta.** GST-tagged *rmdB* was expressed from the pGEX.6p1 vector. His-*whiG* and tag less *rsiG* were expressed from pCOLADuet-1 from two separate multiple cloning sites [33]. Gene expression was induced with 250 µM IPTG and cells were incubated in LB at 16°C overnight. Purification of GST-RmdB, His-WhiG and RsiG was performed using the glutathione sepharose. The eluates were analysed via SDS-PAGE.
(DOCX)

**S6 Fig. GST-tagged GGDEF domain from RmdB interacts with His-WhiG.** GST-*rmdB*GGDEF was expressed with His-*whiG* (A) or alone (B) in *E. coli* BL21 pLysS from two separate plasmids (pIJ10914/His-WhiG and pET15b/GST-RmdBGGDEF). Gene expression was induced with 250 µM IPTG, and cells were incubated in LB at 16°C overnight. Purification of GST-GGDEF~RmdB~ and His-WhiG was performed using the Ni-NTA matrix. Elution was performed using 3x washes with a 50 mM imidazole concentration and eluted with 250 mM imidazole. Eluates were analysed via SDS-PAGE. C. Uncropped gel image. Please note that the three prominent protein band in (A) were cut out from gel and analysed using MALDI-TOF to confirm identity of the relevant proteins.
(DOCX)

**S7 Fig. AlphaFold3 model comparisons of RmdB and RmdB^ΔGGDEF^.** AlphaFold3 [29] was employed to model full-size RmdB and RmdB^ΔGGDEF (Δ258-421aa)^. The structural models are visualised using PyMOL.
(DOCX)

**S8 Fig. Bacterial Adenylate Cyclase Two-Hybrid (BACTH) assays reveal multiple interactions between RmdB and DGCs.** Indicated genes were fused to either the T18 or T25 fragment of the adenylate cyclase from *Bordetella pertussis* on the pUT18/pUT18C or pKT25/pKNT25 vectors that were co-transformed into *E. coli* BTH101. Several clones carrying both plasmids were combined in 1 ml 1xPBS and 10 µl were spotted on LB agar supplemented with 5-Bromo-4chloro-3indolyl-ß-D-galactopyranoside (X-gal; 60 µg/ml), IPTG (0.5 mM) ampicillin (100 µg/ml) and kanamycin (50 µg/ml). Plates were incubated for either 40 hours at 30 °C (A, B) or for 56 hours at 25°C (C, D). Cytosolic fraction of RmdB (Δ1–238aa) was used in BACTH assays as fusing either the T18 or T25 fragment to full size RmdB resulted in protein instability in

 

*E. coli*. Original screening plates are shown on the left (A, C), with each colony carrying a number. Corresponding matrix explaining the identity of each spot is shown on the right (B, D).
(DOCX)

**S1 Table. Strains, plasmids and oligonucleotides used in this study.**
(DOCX)

## Acknowledgments

We thank the research core unit metabolomics at the Hannover Medical School for support with LC-MS/MS analysis and Dr. Susan Schlimpert and Dr. Matt Bush, John Innes Centre, Norwich, for help with the µMACS epitope tag protein isolation procedure.

## Author contributions

**Conceptualization:** Natalia Tschowri.

**Funding acquisition:** Natalia Tschowri.

**Investigation:** Buse Cınar Cakmak, Johanna D. Saric, Neil A. Holmes.

**Methodology:** Buse Cınar Cakmak, Johanna D. Saric, Katrin Wrede, Neil A. Holmes, Julian Haist, Natalia Tschowri.

**Project administration:** Natalia Tschowri.

**Resources:** Natalia Tschowri.

**Supervision:** Mark J. Buttner, Natalia Tschowri.

**Visualization:** Buse Cınar Cakmak, Johanna D. Saric, Neil A. Holmes, Maria A. Schumacher, Natalia Tschowri.

**Writing – original draft:** Buse Cınar Cakmak, Johanna D. Saric, Natalia Tschowri.

**Writing – review & editing:** Buse Cınar Cakmak, Johanna D. Saric, Katrin Wrede, Neil A. Holmes, Maria A. Schumacher, Mark J. Buttner, Natalia Tschowri.

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
