## [Decision Letter · Decision Letter 0]

5 Feb 2026

PGENETICS-D-26-00045

Dual-function enzyme acts as a global c-di-GMP sink and local anti-sigma factor antagonist to drive cellular differentiation

PLOS Genetics

Dear Dr. Tschowri,

Thank you for submitting your manuscript to PLOS Genetics. After careful consideration, we feel that it has merit but does not fully meet PLOS Genetics's publication criteria as it currently stands. Therefore, we invite you to submit a revised version of the manuscript that addresses the points raised during the review process.

We look forward to receiving your revised manuscript.

Kind regards,

Kai Papenfort

Academic Editor

PLOS Genetics

Sean Crosson

Section Editor

PLOS Genetics

Aimée Dudley

Editor-in-Chief

PLOS Genetics

Anne Goriely

Editor-in-Chief

PLOS Genetics

**Additional Editor Comments:**

Dear Dr Tschowri, dear Natalia.

Thank you for submitting your work to PLOS Genetics. The manuscript has now been reviewed by three experts in the field and their comments are provided below. While all referees are highly supportive of your work, they also raised a few major and minor criticisms. We will consider publishing your manuscript if you can address these criticisms in a revised version of the manuscript and your rebuttal letter.

Kind regards,

Kai Papenfort

**Journal Requirements:**

https://journals.plos.org/plosgenetics/s/submission-guidelines#loc-parts-of-a-submission

**Reviewers' comments:**

Reviewer's Responses to Questions

**Comments to the Authors:**

Reviewer #1: This manuscript by Çakmak et al., advances our understanding of how c-di-GMP signalling is terminated to allow developmental progression towards sporulation in Streptomyces venezuelae. The authors identify RmdB as a dual-function phosphodiesterase that integrates global degradation of c-di-GMP with local, protein–protein interactions that directly impact sigma/anti-sigma regulation. These findings are novel and of broad interest to the bacterial signalling and Streptomyces development communities.

The manuscript is clearly written, logically structured and easy to follow. The experimental strategy is well thought out and combines genetics, biochemical assays, protein-protein interaction analyses and phenotyping to back up the manuscript’s conclusions. The use of co-immunoprecipitation, co-elution, and bacterial two-hybrid assays provides complementary and mutually reinforcing support for the proposed model. The identification of RmdB as an interaction hub connecting multiple DGCs further strengthens the idea that spatial and local control of c-di-GMP signalling is central to developmental regulation in Streptomyces venezuelae.

Overall, this is a high-quality study that makes a significant conceptual contribution to the field of signal transduction and developmental biology in Streptomyces venezuelae. I have no concerns or suggestions regarding the experimental design, data quality or interpretation. My only suggestions would be to briefly clarify in the discussion how conserved RmdB is within Streptomyces species and whether other phosphodiesterase-sigma factor interactions have been characterised in any other organisms, and if so, how do these compare to the RmdB-WhiG interaction studied here. These minor suggestions do not detract from the strength of the work presented here.

Reviewer #2: In this study, Çakmak et al explored the role of the Streptomyces phosphodiesterase RmdB in lowering the level of c-di-GMP within cells to allow development to progress. RmdB is a diguanylate cyclase/phosphodiesterase hybrid protein that contains the GGDEF and EAL domains, as well as an additional N-terminal transmembrane domain that anchors the protein to the membrane. The EAL and GGDEF domains are both required for RmdB to function in the timing of developmental progression. Based on the phenotype, the authors performed co-immunoprecipitation and in vivo protein-protein interaction studies using bacterial two-hybrid (BACTH) analysis. These studies revealed that the RmdB GGDEF domain interacts directly with the sigma factor WhiG, thereby assisting in the release from its anti-sigma factor, RsiG. Furthermore, additional BACTH analyses suggest that RmdB interacts with other DGCs, supporting a model in which RmdB acts as a hub that coordinates local interactions and likely the activity of other enzymes involved in c-di-GMP synthesis and degradation.

Overall, this is a well-conducted study with a clear, easy-to-follow set of experiments. The results generally justify the conclusions (see below), though the summary model is preliminary. I believe the interaction studies, but I find it somewhat unsatisfactory that there are so many obvious experiments that the authors are probably planning to perform in the future are not included. How does the GGDEF domain interact with the sigma/anti-sigma complex? Does the interaction affect RmdB activity? Does RmdB binding loosen the interaction between sigma and anti-sigma factors? How does RmdB interact with the other enzymes involved in c-di-GMP synthesis and degradation, and how does this interaction affect their activity? Is there sufficient RmdB to allow binding of all the potential clients? The last paragraph of the Results section is essentially one big question. Given the wealth of data that can be gained from this system, I wonder if this manuscript is a bit premature — but that is not for me to judge.

Questions/remarks

Abstract, line 25 and elsewhere: rather a question – is it common to refer to an anti-sigma factor as ’anti sigma‘?

Introduction: Does the RmdB transmembrane region have a function of its own, e.g., in signal perception?

84, just as a suggestion: The authors may also mention the compartimentalization of c-di-GMP signalling in Caulobacter, where (as in Streptomyces) the cell cycle progression is regulated by this second messenger.

116, end of paragraph, what about RmdA?

134, although the difference appears to be clear for most of the data, some statistics would truly demonstrate the significance.

Figure 3B/C show some cropped data. It would be good, if the non-processed western blots could be added to the supplementary data.

169, Are the authors suggesting that the GGDEF domain binds to the WhiG-RsiG complex via bound c-di-GMP? Does AlphaFold's structure/interaction prediction suggest this? What would a prediction involving the individual proteins (WhiG with RmdB-GGDEF and RsiG with RmdB-GGDEF) look like? Does this support the hypothesis? Does the analysis suggest any critical interaction surfaces or residues that could be substituted?

204, please provide a control with non-His-tagged WhiG

223, how was the stability tested?

Reviewer #3: Cakmak et al. investigated the role of the phosphodiesterase RmdB in c-di-GMP signaling and cell fate decisions in Streptomyces. They report two main findings:

(i) Consistent with their previous study (Haist et al., Mol Microbiol, 2020), the authors demonstrate that RmdB functions as a phosphodiesterase. Accordingly, global c-di-GMP levels are elevated in the rmdB mutant, which correlates with delayed development. Both the EAL domain (essential for phosphodiesterase activity) and the GGDEF domain (typically a diguanylate cyclase) are required for RmdB function in vivo. While this part of the study is technically sound and straightforward, its novelty and mechanistic insight are limited.

(ii) To explore the potential alternative function of the GGDEF domain, the authors propose that RmdB acts as a hub for protein-protein interactions. Pulldown experiments suggest an interaction between RmdB and the sigma factor WhiG, leading to the hypothesis that this interaction antagonizes binding of the anti-sigma factor RsiG to the sigma factor. Unfortunately, the interaction was not supported by the bacterial two-hybrid system. Instead, this assay revealed interactions between RmdB and four diguanylate cyclases. Overall, this second line of research is less well developed, and the available experimental evidence is not yet sufficiently strong enough to substantiate the proposed, appealing model.

Questions and comments:

Determining protein-protein interactions using a single experimental approach carries inherent risks, particularly when a complementary approach (in this case, the BACTH assay) does not support the initial findings. The following questions and suggestions are intended to strengthen confidence in the reported results.

1. Figure 3 lacks information on reproducibility. Does it show one representative example from multiple independent experiments?

2. Figure 3B: The interaction between RmdB and the sigma factor could be validated by a reciprocal pulldown using Glutathione Sepharose.

3. Figure 3B: Given that WhiG-specific antibodies are available (Figure 3D), an informative follow-up experiment would be to recombinantly produce His-tagged RsiG together with native WhiG to test the proposed partner-switching model.

4. Figure 3B: Can the pulldown experiments be repeated using purified proteins? GST- and His-tagged proteins should be easy to obtain and would confirm at least the RmdB-sigma factor interaction.

5. Figure 3C, D: While it is reassuring that the GGDEF variant is present in similar amounts, it remains possible that the protein, which lacks more than 150 amino acids, is not properly folded and therefore unable to engage in proper interactions. Does AlphaFold predict a structured protein or this variant?

6. Can AlphaFold modeling be used to support the proposed RmdB interaction partners?

7. The BACTH experiments were performed using the cytosolic version of RmdB. If the extensive transmembrane region regulates the interactome of the proposed hub protein, it may be worthwhile to fuse the T18 or T25 to the C-terminus of the full-length protein and reassess interactions.

**Have all data underlying the figures and results presented in the manuscript been provided?**

Reviewer #1: Yes

Reviewer #2: **No:** Non-processed western blots should be provided.

Reviewer #3: Yes

PLOS authors have the option to publish the peer review history of their article (what does this mean?). If published, this will include your full peer review and any attached files.

Reviewer #1: **Yes:** Lorena Fernández-Martínez

Reviewer #2: No

Reviewer #3: No

**Figure resubmission:**
---

## [Decision Letter · Decision Letter 1]

8 May 2026

Dear Dr Tschowri,

We are pleased to inform you that your manuscript entitled "Dual-function enzyme acts as a global c-di-GMP sink and local anti sigma factor antagonist to drive cellular differentiation" has been editorially accepted for publication in PLOS Genetics. Congratulations!

Yours sincerely,

Kai Papenfort

Academic Editor

PLOS Genetics

Sean Crosson

Section Editor

PLOS Genetics

Aimée Dudley

Editor-in-Chief

PLOS Genetics

Anne Goriely

Editor-in-Chief

PLOS Genetics

BlueSky: @plos.bsky.social

Comments from the reviewers (if applicable):

Dear Dr Tschowri, dear Natalia.

I have now received feedback from the referees of your revised manuscript and I am happy to inform you that all three suggested to accept it (and so have I). Congratulations on a very nice manuscript.

Best wishes,

Kai Papenfort

Reviewer's Responses to Questions

**Comments to the Authors:**

Reviewer #1: I would like to thank the authors for clarifying my suggestions.

Reviewer #2: The authors have satisfyingly addressed the issues I had with the previous version. Congrats on this beautiful story.

Reviewer #3: The authors have adequately addressed all reviewer comments, conducted additional experiments, and revised the manuscript accordingly. I am satisfied with the revision.

**Have all data underlying the figures and results presented in the manuscript been provided?**

Reviewer #1: Yes

Reviewer #2: Yes

Reviewer #3: Yes

PLOS authors have the option to publish the peer review history of their article (what does this mean?). If published, this will include your full peer review and any attached files.

Reviewer #1: No

Reviewer #2: No

Reviewer #3: No

**Data Deposition**

http://datadryad.org/submit?journalID=pgenetics&manu=PGENETICS-D-26-00045R1

**Press Queries**

---

## [Editor Report · Acceptance letter]

PGENETICS-D-26-00045R1

Dual-function enzyme acts as a global c-di-GMP sink and local anti sigma factor antagonist to drive cellular differentiation

Dear Dr Tschowri,

We are pleased to inform you that your manuscript entitled "Dual-function enzyme acts as a global c-di-GMP sink and local anti sigma factor antagonist to drive cellular differentiation" has been formally accepted for publication in PLOS Genetics! Your manuscript is now with our production department and you will be notified of the publication date in due course.

With kind regards,

Anita Estes

PLOS Genetics

On behalf of:
